# *OsGPAT3* Plays a Critical Role in Anther Wall Programmed Cell Death and Pollen Development in Rice

**DOI:** 10.3390/ijms19124017

**Published:** 2018-12-12

**Authors:** Lianping Sun, Xiaojiao Xiang, Zhengfu Yang, Ping Yu, Xiaoxia Wen, Hong Wang, Adil Abbas, Riaz Muhammad Khan, Yingxin Zhang, Shihua Cheng, Liyong Cao

**Affiliations:** Key Laboratory for Zhejiang Super Rice Research and State Key Laboratory of Rice Biology, China National Rice Research Institute, Hangzhou 310006, China; sunlianping@caas.cn (L.S.); taorangongwangjimm@163.com (X.X.); cnrriyzf@sina.com (Z.Y.); pingping367@163.com (P.Y.); 18883966700@163.com (X.W.); wjiyinh@126.com (H.W.); adilabbasqau@126.com (A.A.); riazkatlang@126.com (R.M.K.); zyxrice@163.com (Y.Z.)

**Keywords:** anther wall, tapetum, pollen accumulation, *OsGPAT3*, rice

## Abstract

In flowering plants, ideal male reproductive development requires the systematic coordination of various processes, in which timely differentiation and degradation of the anther wall, especially the tapetum, is essential for both pollen formation and anther dehiscence. Here, we show that *OsGPAT3*, a conserved glycerol-3-phosphate acyltransferase gene, plays a critical role in regulating anther wall degradation and pollen exine formation. The *gpat3-2* mutant had defective synthesis of Ubisch bodies, delayed programmed cell death (PCD) of the inner three anther layers, and abnormal degradation of micropores/pollen grains, resulting in failure of pollen maturation and complete male sterility. Complementation and clustered regularly interspaced short palindromic repeats (CRISPR)/CRISPR-associated 9 (Cas9) experiments demonstrated that *OsGPAT3* is responsible for the male sterility phenotype. Furthermore, the expression level of tapetal PCD-related and nutrient metabolism-related genes changed significantly in the *gpat3-2* anthers. Based on these genetic and cytological analyses, *OsGPAT3* is proposed to coordinate the differentiation and degradation of the anther wall and pollen grains in addition to regulating lipid biosynthesis. This study provides insights for understanding the function of *GPATs* in regulating rice male reproductive development, and also lays a theoretical basis for hybrid rice breeding.

## 1. Introduction

Rice is a key gramineous plant that is self-pollinated. Guaranteeing sufficient rice yield requires stable male sterility, which depends on normal development of anther and male gametophyte (pollen) formation [1,2]. Typical rice anthers have four lobes, and the central reproductive microsporocytes (or pollen mother cells) are surrounded by four concentrically organized somatic cell layers in each locule: the epidermis, endothecium, middle layer, and tapetum. Anther development is a multistage process involving localized cellular differentiation and degeneration, cell division and chromosomal behaviors, and synthesis and transportation of nutrients. This process is combined with changes in the structure and external environment to complete anther dehiscence and pollen maturation, and release for pollination and fertilization. All four cell layers possess specific functions and coordinate throughout the whole process to ensure normal anther development and microspore/pollen formation in rice [2,3,4,5,6]. The epidermis is located at the outermost layer of anthers to protect against external environmental stresses to ensure normal development of internal cells at the appropriate time for anther dehiscence and pollen release. The endothecium, with localized secondary thickening, is the second layer and is essential for anther dehiscence and pollen release. Endothecial development is concurrent with pollen maturation and degeneration of the anther tapetum and middle layer. The middle layer, which is situated between the tapetum and endothecium, undergoes programmed degeneration along with the tapetum during the pollen maturation stage [5,6,7,8]. The innermost cell layer of the anther wall, the tapetum, directly contacts developing gametophytes. The tapetum undergoes programmed cell death (PCD)-mediated degeneration to supply a series of nutritional components and structural molecules for normal pollen formation and ordinary anther development [9,10]. Proper development and timely degeneration of tapetal cells is essential for providing and supplying nutrients for sporopollenin synthesis and pollen development [1,11,12,13]. All developmental processes of these cell layers are dominated by specific regulators with extremely precise and systematic molecular mechanisms; mutation of these genes may result in anther development malformations and eventually lead to pollen abortion [14,15].

Programmed cell death (PCD) events, known as apoptosis, are often characterized by nuclear chromatin condensation and degeneration, membrane breakdown, and compactness of cytoplasmic organelles. PCD is usually used as a cytological feature of tapetum degradation in plants [9,12,16]. The PCD process also occurs in the most peripheral layers at the late stages of anther development. Vascular bundle cells, the cavity of dehiscence, the epidermis, and the endothecium are always prepared for apoptosis to provide nutrients for pollen mitosis and maturation [17]. Timely PCD of the anther wall, especially tapetum degradation in vivo, plays a fundamental role in male reproduction. Premature or delayed tapetal PCD and cellular degeneration can cause pollen abortion and male sterility [2,9,12,15]. Recently, multiple genes were identified to play essential roles in the process of rice tapetal PCD and pollen development, including several basic helix–loop–helix (*bHLH*) transcription factors, undeveloped tapetum 1 (*UDT1*, *bHLH164*) [11], tapetum degeneration retardation (*TDR*, *bHLH5*) [12,18], eternal tapetum 1/delayed tapetum degeneration (*EAT1*/*DTD*, *bHLH141*) [19,20], TDR-interacting protein 2 (*TIP2*, *bHLH142*) [21,22,23,24], MYB family transcription factor *GAMYB* [25,26,27], PHD-finger protein persistent tapetal cell 1 (*PTC1*) [28], and TGA transcription factor *OsTGA10* [29]. These genes were shown to regulate various aspects of anther development, especially tapetal PCD. *GAMYB* probably works upstream of *TDR1* and *PTC1* in parallel with *UDT1* to regulate rice tapetum development and pollen wall formation [27,28]. *TIP2* functions upstream of *TDR* and *EAT1*, but downstream of *UDT1*, and also binds to the promoter of *EAT1* to activate positive effects on regulation of tapetum PCD by promoting aspartic proteases *AP25*, *AP37*, and *OsCP1* [20,22]. *OsTGA10* also regulates tapetum development and pollen formation by interacting with *TIP2* and *TDR* to affect the expression of *AP25* and *MTR* [29]. In addition, some genes that encode enzymes or specific binding proteins are also essential for tapetal degeneration. Loss of function of these genes can result in anther deformity and defects in pollen exine formation. *DTC1* controls tapetal degeneration by modulating the dynamics of reactive oxygen species (ROS) with *OsMT2b* during reproduction of male rice [30]. *DEX1* regulates tapetal cell death and pollen exine formation by binding to Ca^2+^ to modulate cellular Ca^2+^ homeostasis, acting as a component required for tapetal cell death signal transduction [31]. Recessive mutation of the fasciclin glycoprotein *MTR* can dislocate its plasma membrane localization system and cause delayed tapetum PCD and reduced synthesis of Ubisch bodies, which are micron-sized particles on the inner surface of the tapetum in anthers, finally resulting in abortive pollen grains and complete male sterility [32]. The F-box protein *OsADF* is expressed in tapetal cells and microspores and works depending on *TDR* by binding E-box motifs of its promoter [33]. The two aspartic proteases, *AP25* and *AP37*, cysteine protease *OsCP1*, and apoptosis inhibitor *API5* are reported as specific regulators of the tapetal PCD process, and inhibition/mutagenesis of these genes can cause defects in pollen formation and eventually lead to male sterility [12,20,34].

Microspore cells need to undergo a series of processes after they are released from the tetrads to ultimately develop to mature fertile pollen grains, such as pollen wall formation, vacuolation, two rounds of mitosis, and starch enrichment. Normal progress of these processes requires nutritional supply and structural support from aliphatic biopolymers (sporopollenins, Ubisch bodies, epicuticular waxes, and cuticle monomers) and especially the synthesis and transport of sporopollenin. Sporopollenin is made up of polyhydroxylated aliphatic compounds and oxygenated aromatic monomers, such as phenolics, conjugated by ether and ester bonds, and acts as one of the main components of pollen exine, playing a critical role during pollen development by protecting pollen grains from abiotic and biotic stresses. Evidence from previous studies of male-sterile mutants with abnormal pollen wall formation showed that many genes are involved in these biosynthesis/transport mechanisms [1,2,15,35,36]. Two cytochrome P450 family genes, *CYP704B2* and *CYP703A3*, function as catalyzers of ω-hydroxylated fatty acids with 16- and 18-carbon chains and in-chain hydroxylase only for lauric acid to generate 7-hydroxylated lauric acid [37,38,39]. Defective pollen wall (*DPW*) acts as a fatty acyl-carrier protein reductase and *DPW2* as a fatty-acid acyltransferase to alter the amounts of cutin and waxes, and of lipidic and phenolic compounds, respectively, during anther development and pollen formation [40,41]. The ATP-binding cassette (ABC) transporters *ABCG15*/*PDA1*, *ABCG26*, and *ABCG3* work collaboratively but perform their own functions to transport different materials for anther development and pollen formation [42,43,44,45]. The lipid transfer protein *OsC6* is secreted into anther cuticle, anther locule, and the space between the tapetum and middle layer for pollen exine and orbicule formation downstream of *TDR* and *GAMYB* [46]. In addition, *WDA1* [47], *OsACOS12* [48,49], *OsNP1* [50], and *OsPKS2* [51,52] were also reported to regulate sporopollenin biosynthesis and deposition. Loss of function of these genes caused defects in not only the anther cuticle and pollen wall, but also the number of secretory, lipidic Ubisch bodies. In fact, the functions of these genes often interact; most of the recessive mutants of PCD-induced genes showed apparent defects in Ubisch body patterning and pollen exine formation. In particular, *tdr*, *ptc1*, *eat1-1*, *dex1*, and *dtc1* mutants exhibit obvious microspore collapse and pollen wall degeneration at the microspore stage [12,20,28,30]; *CYP703A3*, *ABCG15*, *ABCG26*, and *OsACOS12* also correspondingly showed high expression levels in tapetal cells and their recessive mutants due to delayed or premature degradation of the tapetum layer [42,43,44,49].

Acyl coenzyme A (CoA) glycerol-3-phosphate acyltransferases (GPATs), which localize to the endoplasmic reticulum (ER), are generally recognized as important catalyzers for the first step of de novo synthesis of triacylglycerol. GPATs play key roles in regulating cell growth and metabolic processes of membrane lipids, storage lipids, and extracellular lipid polyesters (cutin and suberin) by generating lysophosphatidic acids (LPAs) and acylating glycerol 3-phosphate at the sn-1 or sn-2 hydroxyl with acyl-CoA or acyl-acyl-acyl carrier protein (ACP) to alter glycerolipid triacylglycerol (TAG) biosynthesis [53,54,55]. The sn-1 GPATs promote acylation to produce lysophosphatidic acid (LPA) for lipid formation, while sn-2 GPATs possess a phosphatase domain to produce sn-2 monoacylglycerol (2-MAG) as the major product for cutin and suberin synthesis in plants. Most GPATs were confirmed to have sn-2 acyl transfer activity [56,57,58,59,60]. In *Arabidopsis*, there are ten GPATs, eight of which (*AtGPAT1* to *AtGPAT8*) belong to the sn-2 family and are divided into three sub-clades. The first clade, *AtGPAT1* to *AtGPA3*, is mainly expressed in flowers and siliques, and show sn-2 acyltransferase during the process of dicarboxylic acyl-CoA substrate utilization but not phosphatase activity. *Arabidopsis gpat1* mutant exhibits less fibrillar material, fewer vesicles in the anther locule, disrupted degeneration of the tapetum, and collapsed pollen grains [61]. *AtGPAT4*, *AtGPA6*, and *AtGPAT8* are unique bifunctional enzymes with both sn-2 acyltransferase and phosphatase activity to produce 2-MAG for cutin synthesis. *AtGPAT5* and *AtGPAt7*, which were identified as part of the suberin-associated clade, function in suberin synthesis in the wounding response for root and seed coat formation [60]. Genetic functions revealed that *AtGPAT1* and *AtGPAT6* play an important role in anther development and pollen formation. Loss of function of these two genes can result in altered ER profiles in tapetal cells, reduced pollen production, and decreased pollination; double mutation of these two genes can cause defective callose degeneration, pollen release, and complete male sterility [60,61]. Homologous sequence alignments and phenotypic analysis on other species reveal that *SlGPAT6*, *OsGPAT3*, and *ZmGPAT3* also have important roles in regulating anther cuticle biosynthesis and pollen exine formation [62,63,64]. However, studies on the metabolism of GPATs in regulating rice male reproductive development are still limited, particularly in tapetum PCD, and there is a lack of related mutants and phenotype analysis of relevant genes.

In this study, we further characterized *OsGPAT3* in rice male reproductive development, particularly focusing on its function in anther wall degeneration and pollen maturation. The *gpat3-2* mutant exhibited delayed tapetum PCD and Ubisch body formation and abnormal anther wall and pollen degeneration, resulting in complete male sterility. Genetic analysis and map-based cloning revealed that the mutant phenotype was caused by a single-nucleotide polymorphism (SNP) mutation in the first exon of *OsGPAT3*, a land plant sn-2 *GPAT* homolog. Two allelic mutants from our mutant library and three other mutant alleles of *OsGPAT3* generated using clustered regularly interspaced short palindromic repeats (CRISPR)/CRISPR-associated 9 (Cas9) also showed the same male-sterile phenotype. In addition, the expression pattern of many tapetum PCD-induced regulators and nutrition metabolism related genes were significantly altered resulting from recessive mutation of *OsGPAT3*. Thus, our results demonstrate that *OsGPAT3* is essential for anther wall PCD and pollen development in addition to its function during the synthesis of Ubisch bodies for anther cuticle and pollen exine formation. Our study also provides new insights into GPATs on regulating tapetum PCD and pollen maturation during plant reproductive development.

## 2. Results

### 2.1. Isolation and Phenotypic Analyses of the gpat3-2 Mutant

The *gpat3-2* mutant was first identified from the ethyl methyl sulfone (EMS)-soaked M1 progeny of an *indica* rice cultivar Zhonghui8015 (Zh8015) in Lingshui, Hainan Province. The *gpat3-2* mutant exhibited complete male sterility and was genetically stable in Hangzhou, Zhejiang Province. When the *gpat3-2* mutant was pollinated with wild-type pollens, all BC_1_F_1_ plants exhibited normal male fertility. Further identification of anthers in the BC_1_F_2_ population also indicated that the *gpat3-2* phenotype conformed to the genetic regulation of a single recessive gene (χ^2^ = 0.17; *p* < 0.05). Genetic analysis on the F_1_ and F_2_ progeny generated by the cross *gpat3-2* × 02428, a wide compatibility *japonica* cultivar, also confirmed the monofactorial recessive inheritance of the *gpat3-2* mutant (see Appendix A, Appendix A).

Vegetative development, including plant height, heading date, tiller number, other major agronomic traits, and general spikelet morphology of the *gpat3-2* plants did not differ from those of wild-type Zh8015 except for the anthers (Figure 1A–C). The Zh8015 anthers were normal golden yellow and had countless fertile pollen grains in the anther interior (Figure 1C–F); anther dehiscence occurred soon after glumes opened and pollens were deposited for fertilization (Figure 1C,D). Compared with wild-type anthers, the mutant anthers were small and white, without mature pollen grains (Figure 1C–F,H) and failed to dehisce, resulting in completely sterile spikelets (Figure 1C–F,H).

### 2.2. Defects of Anther Development and Pollen Maturation in gpat3-2 Mutant

To identify defects in the *gpat3-2* mutant, semi-thin transverse sections of anthers at different developmental stages from the wild-type plant and *gpat3-2* mutant were further examined. Rice anther development was delineated into 14 stages, which was consistent with that of *Arabidopsis thaliana* [3], based on the cellular events observed under light microscopy by semi-thin section [65,66]. By stage 7, both wild-type (WT) and *gpat3-2* anthers exhibited an obvious four-layered anther wall from surface to interior, and the wall enwrapped pollen mother cells (PMCs) within the locule. At this stage, the PMCs progressively initiated meiotic division and nestled against the tapetal layer. No obvious defects in the four somatic layers of the anther wall and microsporocytes were detected between the WT and *gpat3-2* mutant until this stage (Figure 2A,B). Subsequently, *gpat3-2* anthers began displaying obvious morphological abnormalities. At late s7 to s8a, the PMCs generated dyed cells, the middle layer became nearly invisible, and the anther wall had three layers in WT anthers. Meanwhile, the tapetal layer began having a weak point of programmed cell death (PCD) (Figure 2C). In the *gpat3-2* anthers, the middle layer was still clearly visible, the anther wall had the four-layer structure, and newly formed dyed cells were misshapen and less darkly stained; this phenomenon was also observed in tapetum cells (Figure 2D). By the end of stage 8b, tetrads were generated, tapetum PCD started, and the middle layer became nearly invisible in the wild-type anthers (Figure 2E). The *gpat3-2* anthers also formed imperfect tetrads, which exhibited unequal cleavage. However, the tapetum layer was electron-dense and expanded, and the middle layer was still apparent and showed no signs of degradation (Figure 2F).

At stage 9, microspores were released from the tetrads and tapetal cells, becoming condensed and electron-dense resulting from gradual PCD-induced degradation (Figure 2G). Although microspore cells can be formed after meiosis, most anther walls gradually became distorted and shrank, and the middle layer remained relatively discernible in this period in the mutant anthers. Furthermore, the *gpat3-2* mutant swelled and there were lightly stained tapetal cells, indicating abnormal PCD (Figure 2H). A sharp distinction between the wild-type and *gpat3-2* anthers began appearing at stage 10; normal vacuolated microspores were uniformly attached to the tapetum side as round shapes with dark-stained pollen exine, while the tapetum layer was electron-lucent after gradual PCD (Figure 2I). By comparison, internal cavities of the *gpat3-2* anthers were disordered; the tapetum layer was swollen and lightly stained with obvious degradation characteristics and the middle layer was still visible. In addition, microspores were disrupted and degraded together with the tapetum layer (Figure 2I). From the pollen mitosis stage and the mature pollen stage, wild-type pollen formed a complete double-layer exine and fertile pollen after two mitotic divisions with pollen exine deposition and starch accumulation, while the tapetum layer gradually degenerated and thinned until it almost disappeared at the end of stage 12. During this procedure, the epidermis and endothecium layer further degenerated and anther dehiscence occurred. Mature pollen grains were full of lipids, starch, and other storage nutrients, and were dark-stained with toluidine blue, indicating that the wild-type pollen grains had normal functions and were viable (Figure 2K,M,O). However, degradation of microspores and tapetal cells continued, resulting in linear pollen walls and cell detritus in the *gpat3-2* anther locule. In addition, the outer layers of the anther wall retained the original structure; epidermis, endothecium layer, and middle layer were not degraded until the end of stage 13 in *gpat3-2* anthers (Figure 2L,N,P). These observations suggested that *gpat3-2* carried defects not only in anther wall development and pollen maturation, but also in differentiation and degradation of tapetal cells and anther wall.

### 2.3. Delayed PCD of Osgpat3-2 Tapetal Cells and Anther Wall Cells

The transverse section analysis suggested that *gpat3-2* mutation affected the differentiation and degradation of tapetal cells and anther wall. We, therefore, used a terminal deoxynucleotidyl transferase-mediated deoxyuridine triphosphate (dUTP) nick-end labeling (TUNEL) assay to test the tapetum PCD process during a range of developmental stages in WT and *gpat3-2* anthers.

Microspore mother cells were generated during meiosis and tapetal cells became condensed after meiosis; there was no detectable DNA fragmentation signal in the wild-type or *gpat3-2* tapetal cells. At stage 8a, some TUNEL-positive nuclei were detected in wild-type tapetal cells and the middle layer, indicating that normal PCD started occurring in the wild-type anthers (Figure 3A). At stage 8b, positive signals of DNA fragmentation were much stronger in wild-type tapetal cells. Positive PCD signals were also detected in both the outer layers (endothecium and middle layer) and vascular bundle cells of wild-type anthers at this stage (Figure 3C). However, no visible fragmented DNA signal was observed in the *gpat3-2* mutant anthers at both stage 8a and 8b (Figure 3B,D). At stage 9, when the microspore was released from the tetrad, PCD signals in wild-type tapetal cells became strongest, and the signal in the outer layers, vascular bundle cells, and cavity of dehiscence increased at the same time (Figure 3E). However, PCD signals were still not detected in the *gpat3-2* tapetum; only weak signals of DNA fragmentation were detected in *gpat3-2* outer layers and vascular bundle cells (Figure 3F). At stage 10, positive PCD signals in wild-type tapetal cells and other tissues became much weaker than at stage 9 (Figure 3G), while PCD signals unexpectedly became detectable and very strong in most tissues of *gpat3-2* anthers, including tapetal cells, the outer layers (endothecium and middle layer), vascular bundle cells, and even microspore cells inside the chamber (Figure 3H). At stage 12, PCD signals of wild-type tapetal cells gradually became invisible and only small, positive signals were detected in the outer layers and cavity of dehiscence (Figure 3I). Conversely, PCD signals continued increasing in all parts of *gpat3-2* anthers (Figure 3J). Nevertheless, at the dehiscence stage, none positive PCD signals were detected in wild-type tapetum, but the signals became much stronger in outer layers, vascular bundle cells, and cavity of dehiscence to meet the requirement of anther cracking and pollination (Figure 3K). However, positive PCD signals became strongest in all cell layers of *gpat3-2* anthers at this stage, including the degenerated microspores. These TUNEL assays demonstrate that the PCD of tapetum and peripheral layers were normal and orderly, and the delay and disorder of PCD in tapetum, anther wall, and microspores possibly resulted in the failure of pollen formation in the *gpat3-2* mutant.

### 2.4. Osgpat3-2 Exhibits Defects in the Formation of Ubisch Bodies and Pollen Accumulation

For further verification and understanding of abnormalities of the *gpat3-2* mutation, we performed a more detailed scanning electron microscope (SEM) observation of the surfaces of wild-type and *gpat3-2* anthers and pollen grains at different development stages. No obvious differences of the anther morphologies or outer wall structure were observed between wild-type and the *gpat3-2* anthers at stage 9; both showed a smooth epidermis and microspores were released (Figure 4A,B,D-I,D-II). However, further enlargement of the inner surface of locules and pollen exine revealed that the Ubisch bodies were produced by the tapetum and transported to microspores in wild-type anthers, while the *gpat3-2* anthers still had a smooth inner surface and abnormal pollen exine, indicating the defects of tapetum PCD (Figure 4C,E-I,E-II). From the vacuolated pollen stage (stage 10) to the mature pollen stage (stage 12), the size of wild-type anthers gradually increased to almost double, the anther epidermis constantly thickened and was covered with a three-dimensional spaghetti-like cutin layer, the continuous synthesis and transport of Ubisch bodies resulted in a neat arrangement of inner surface, and the uninucleate microspores enlarged and developed to trinucleate pollen with regular shaped pollen exine, which was formed by secreted tapetum-produced sporopollenin precursors from Ubisch bodies (Figure 4III,V,VII of A–E). By contrast, the size of *gpat3-2* anthers was only slightly increased from stage 9 to 11 and basically no longer increased after this stage (Figure 4A-II,IV,VI,VIII). Anther epidermis was still smooth, which indicated defects in the synthesis of typical cutin and in the synthesis of fatty acids in the anther wall (Figure 4B-II,IV,VI,VIII). The inner wall of anthers gradually became irregular at stages 11 and 12 (Figure 4C-VI,VIII) and was smooth but uneven with randomly distributed flocs of Ubisch bodies and degraded microspores at stages 9 and 10 (Figure 4B-II,IV). Further observation of the microspores from different developmental periods showed that microspores in *gpat3-2* anther locules were gradually collapsed and covered with randomly distributed cutin materials, finally leading to the *gpat3-2* microspores becoming more and more chaotic-like cotton wools (Figure 4E-II,IV,VI,VIII).

Transmission electron microscopy (TEM) observation verified the defects in *gpat3-2* anthers and pollen grains. Obvious differences were first identified at stage 8a; *gpat3-2* anthers showed less tapetum differentiation but more vacuoles and a thicker middle layer (see Appendix A, Appendix A) and pollen mother cells were agglomerated without nucleus compared with the wild-type anther (see Appendix A, Appendix A). At stage 8b, tapetal cells of wild-type anthers began differentiating and became sparse and loose (see Appendix A, Appendix A), while the *gpat3-2* tapetum cells were still electron-dense without any signs of degradation (see Appendix A, Appendix A). In addition, tetrads of WT were first surrounded by callose that resembled the pollen primexine structure (see Appendix A, Appendix A); however, tetrads of *gpat3-2* were still naked without a wrapping layer as before (see Appendix A, Appendix A). These results were almost the same as previously reported by Men et al. [61]. However, at stage 9, the tapetum gradually turned into a small, loose thread resulting from the vigorous PCD process, and only a very small line of residual cells of the middle layer was observed in wild-type anthers (Figure 5A-1,C-1,E-1). Ubisch bodies, which were believed to secrete tapetum-produced sporopollenin precursors for pollen exine formation, began being released from the tapetum and gradually grew into electron-dense orbicules (Figure 5E,F). Pollen exine of microspores with accumulation of particulate sporopollenin in the wild-type locule was visible at this stage (Figure 5G-1,I-1). Although microspores and pre-Ubisch bodies seemed to be formed from the mutant tapetum at stage 9, the tapetal cells and microspores were still electron-dense without any signs of differentiation (Figure 5B-1,D-1,F-1,H-1). In addition, some accumulation of electron-dense materials occurred on the outer surface of mutant microspores, but these structure were unable to grow into mature Ubisch bodies, and formed an abnormal exine structure (Figure 5J-1). From stage 10 to stage 11, the middle layer disappeared completely, and the endothecium layer constantly degraded and thinned accompanied by the collapse of epidermis (Figure 5A-2,C-2,A-3,C-3); the tapetum layer differentiated and degenerated to a lightly stained and thin layer (Figure 5A-2,C-2,E-2,A-3,C-3,E-3). Microspore exine deposition, which underwent vacuolization and two rounds of mitosis, was completed and eventually formed thick exine with distinctive layers of tectum, bacula, and nexine under normal nutrient supply from continuously generated Ubisch bodies (Figure 5G-2,I-2,G-3,I-3). Although the tapetum of the *gpat3-2* mutant appeared to be degraded, it was still dense and deeply stained (Figure 5B-2,D-2,F-2,B-3,D-3,F-3): the precursor of newly formed pre-Ubisch bodies failed to synthesize complete and mature Ubisch bodies (Figure 5F-2,F-3). In addition, the epidermis and endothecium cells of anthers were deformed, but the middle layer was still obviously visible and thick (Figure 5B-2,D-2,B-3,D-3). The microspores formed previously in *gpat3-2* anthers exhibited irregular pollen exine and were also gradually degraded with tapetal cells, resulting in degraded cell remnants in the anther locule (Figure 5B-2,H-2,J-2,B-3,H-3,J-3). At the mature pollen stage (stage 13), a distorted layer of epidermis was also visible in wild-type anther, and the endothecium cells and tapetum were almost degraded with numerous Ubisch bodies attached to the inner side facing the pollen grains (Figure 5A-4,C-4). At stage 13, wild-type microspores grew to spherical pollen grains full of starch, lipids, and other nutrients surrounded by a normal bilayer exine (Figure 5G-4,I-4, white arrowhead). However, the anther wall of the *gpat3-2* mutant was still a three-layer outer structure with a few lipids deposited on the inner side of the middle layer; the tapetum was prominently electron-lucent and no longer a distinct layer at this time, probably due to serious degradation (Figure 5B-4,D-4,F-4). The *gpat3-2* microspores aborted and collapsed, and the framework of the exine distorted and folded without the internal nexine layers; only remnants of abnormal epidermis in the locule were left, resulting from severe degradation (Figure 5H-4,J-4). Ultimately, all the defective tissue burst out; finally, the anther chamber of the *gpat3-2* mutant was filled with degraded fragments and residual abnormal pollen grains.

Together, all these results indicated that the *gpat3-2* mutation affected the differentiation and degradation of the anther wall, the synthesis and supply of Ubisch bodies, and pollen wall formation. The *gpat3-2* defects were different from the *gpat3* mutant reported previously, because *gpat3* microspores could not be released from the tetrads and were still covered with callose at the young microspore stage [63].

### 2.5. Fine Genetic Mapping and Candidate Gene Analysis of the Osgpat3-2 Mutation

To map the *gpat3-2* gene, map-based cloning was performed using the F_2_ population from the cross between *gpat3-2* and 02428. Polymorphisms were confirmed as described previously [39,67]. Seven individuals each of the wild-type and *gpat3-2* phenotypes were chosen for linkage analysis using the bulked segregant analysis (BSA) method and the results revealed that *gpat3-2* is located on the long arm of chromosome 11, flanked by simple sequence repeat (SSR) loci RM27172 and RM27326 (Figure 6A). Further primary mapping was conducted using encryption markup with an insertion/deletion (InDel) marker RD1110 and an SSR marker RM27273 in 176 F_2_ recessive individuals; the mutation locus was in a 871.4-kb region between markers RM17273 and RM27326 (Figure 6A). For fine-scale mapping of the *gpat3-2* gene, five significantly polymorphic InDel markers were designed based on polymorphisms between *japonica* Nipponbare and *indica* 9311 [68] (Appendix A, Appendix A). Using these newly developed markers along with high-resolution genetic linkage analysis with 1354 recessive individuals from F_2_ populations of the cross between *gpat3-2* and 02428, the *GPAT3-2* gene was finally delineated to a 26-kb region between ZH-3 and ZH-6 (Figure 6B).

We sequenced the three open reading frames (ORFs) in this region according to the Rice Genome Annotation Project (http://rice.plantbiology.msu.edu/index.shtml) (Figure 6C; Appendix A, Appendix A) of both the WT and *gpat3-2* mutant. We found that the *gpat3-2* mutant carried a single nucleotide mutation (G to A) in the first exon of *Loc_Os11g45400* (Figure 6D–F), which resulted in the corresponding amino acids being mutated directly from tryptophan to a stop codon (Figure 6D,G). Two pairs of calcium-dependent activator protein for secretion 1 (CAPS1) enzyme digestion primers were designed to test and verify this alternative splicing site for this mutation using the endonuclease *Nla*IV, and the enzyme digestion results confirmed our prediction (Figure 6H).

### 2.6. Function Verification of the OsGPAT3 Gene

To confirm that male sterility was caused by the mutation in *Loc_Os11g45400*, a 7.9-kb *OsGPAT3* genomic fragment (*gOsGPAT3*) was transformed into calli induced from young panicles of homozygous *gpat3-2* mutant plants to rescue the sterile phenotype of *gpat3-2* (Figure 7A). The *gOsGPAT3* fragment included the 3.35-kb upstream sequence, the full-length ORF of *Loc_Os11g45400* from the wild type, and a 1180-bp region downstream from the termination codon sequence. The transgenic positive plants had a normal seed-setting rate, similar to those of wild type (Figure 7B). The complemented lines exhibited a normal seed-setting rate (Figure 7B) and golden yellow anthers (Figure 7C), and pollen grains accumulated abundant starch granules and could be dyed black by 1.2% I_2_/KI solution (Figure 7D–F). In addition, anther transverse sections were examined to further verify the anther developmental process in complementary lines. The microspore mother cells secreted normal microspores at stage 9, the middle layer was invisible, and tapetal cells degraded and had fine lines and irregular shapes (Figure 7G). At stage 11, the microspores underwent mitotic divisions and generated binucleate pollen grains, while the endothecium layer and tapetum cells gradually tapered off and epidermis cells were puffy (Figure 7H). Normal developmental morphologies were also visible at stage 13; fertile pollen grains were produced, the epidermis layer was almost all that was left at this stage, and the anther chamber started shrinking and bursting for pollination (Figure 7I). Therefore, the whole developmental process of the anther in complementation lines was restored. These experiments confirmed that the single-nucleotide mutation in *Loc_Os11g45400* was responsible for the no-pollen phenotype of *gpat3-2*.

To further confirm this result, we designed a target within *Loc_Os11g45400* using CRISPR/Cas9 in the Zh8015 genetic background and obtained three other ideal mutants, *gc-1*, *gc-2*, and *gc-3* (Figure 8A). We also found two allelic mutants, one with a single-base insertion (*gpat3-3*) and the other with a single-nucleotide mutation (*gpat3-4*) (Figure 8A). As expected, these five mutants had small, white anthers without mature pollen grains and with significantly increased transcription levels similar to *gpat3-2* (Figure 8B,C–H,O). A TUNEL assay also uniformly observed disordered anther locule with significantly enhanced PCD signals in almost all layers of anther wall, vascular bundle cells, and cavity of dehiscence compared to wild-type anther (Figure 8I–N). As predicted by the Rice Genome Annotation Project, OsGPAT3 protein includes a typical signal peptide, two transmembrane regions, and a conserved GPAT domain, which contains four acyltransferase motifs, without a phosphatase domain (see Appendix A, Appendix A). Previous studies showed that OsGPAT3 belongs to the first clade of the conserved land plant sn-2 GPAT family that specifically regulates lipid biosynthesis for anther cuticle and pollen exine formation [60,61,62,63,64]. Further multiple comparisons of the *OsGPAT3* sequence found that the *gpat3-2* mutant and all three CRISPR/Cas9-induced mutants had premature stop codons and produced truncated polypeptides, which resulted in destruction of the conserved domains. While the two allelic mutants were somewhat different, the protein structure of *gpat3-3* changed completely after the single-base insertion and *gpat3-4* carried an amino-acid conversion in the most conservative GPAT domain (see Appendix A, Appendix A). The findings indicated that the six mutant lines carried different mutations which disrupted the function of the conserved GPAT domain, transmembrane region, and acyltransferase motifs (Figure 7; Appendix A, Appendix A). Taken together, the results demonstrated the function of *Loc_Os11g45400* in rice anther wall PCD and pollen development.

### 2.7. Mutation in OsGPAT3 Affects the Expression of Genes Involved in Both Tapetum PCD and Nutrient Metabolism

*OsGPAT3* was reported to regulate the biosynthesis of lipid metabolism for anther cuticle and pollen exine formation [63]. Our results indicated that the mutation of *OsGPAT3* caused severely delayed tapetum PCD and anther wall development, failure of Ubisch body formation, and abnormal degradation of microspores. To understand the role of *OsGPAT3* in lipid metabolism, anther wall development, and tapetum PCD, and to explain the phenotype in *gpat3-2*, we compared the expression level of a series of genes regulating tapetum PCD and synthesis/transportation of sporopollenin precursors during male reproductive development between the wild type and *gpat3-2* mutant using qPCR. The expression pattern of *OsGPAT3* is consistent with the defects shown in *gpat3-2* mutant and is exactly like *PTC1* (Figure 9A, Li et al.) [28]. The *gpat3-2* anthers showed higher expression than the wild type at stage 7, lower expression during meiosis, and tetrad formation (stage 8), but significantly higher expression than the wild type after stage 9, when young microspores were released (Figure 7A). Nearly half of the tapetum PCD-related genes were downregulated in the *gpat3-2* mutant, including *GAMYB*, *DEX1*, and three cysteine proteases, *AP25*, *AP37*, and *OsCP1* (Figure 9A). Eight genes, including two homologous genes of *OsGPAT3* in rice (*Loc_Os05g38350* and *Loc_Os10g42720*), four bHLH transcription factors (*UDT1*, *TDR*, *EAT1*, and *TIP2*) that particularly regulate tapetum degeneration, *OsTGA10*, which is the target transcription factor of *OsMADS8*, and *MTR*, showed coincident upregulation in the *gpat3-2* mutant (Figure 9A). This may further prove that the organelles are still active and the metabolic process was still occurring during late stages of the *gpat3-2* mutant.

The expression pattern of genes involved in lipid/carbohydrate metabolism, synthesis/transportation of sporopollenin precursors and Ubisch bodies, and pollen wall formation also had some interesting features. The majority of genes related to anther cutin biosynthesis and pollen exine formation had significantly reduced expression in the *gpat3-2* mutant, including *CYP704B2*, *CYP703A3*, *OsACOS12*, *DPW*, *DPW2*, *OsC6*, *WDA1*, and *OsNP1*, further confirming the metabolic disorders of these nutrients in the *gpat3-2* mutant (Figure 9B). Three glycometabolism-related regulators, *CSA*, *UGP1*, and *UGP2*, also showed significant downregulation in the *gpat3-2* mutant, indicating that distortion of many other metabolic processes may concomitantly occur in the mutant (Figure 9B). However, the expression levels of two genes were significantly increased in the *gpat3-2* mutant: *MSP1*, a Leu-rich repeat receptor-like protein kinase that affects the number of cells entering into sporogenesis while simultaneously initiating anther wall formation in rice [69], and *OsMADS3*, which regulates late anther development and pollen formation by modulating ROS homeostasis [70]. The expression pattern of two ATP-binding cassette G transporters, *OsABCG15/PDA1* and *OsABCG26*, showed the same upregulation at stage 8 and mature pollen stage (stage 12), but unexpectedly had different expression patterns at stage 8 to stage 10 (Figure 9B). The transcriptional level of *OsABCG15/PDA1* in *gpat3-2* anthers was continuously higher than that of the wild type; however, *OsABCG26* was evidently downregulated at these stages. Together, these results suggest that loss of function of the *OsGPAT3* mutation may affect not only the biosynthesis of lipidic compounds and the anther wall PCD process, but also the formation/transport of Ubisch bodies and sporopollenin precursors, all of which are essential for anther development and pollen formation.

## 3. Discussion

Rice male reproductive development is determined by a series of specific functional factors, and loss of function of these genes may result in malformed anthers and eventually cause pollen abortion and male sterility, which are essential for hybrid rice breeding. Elucidation of the genetic, molecular, and biochemical mechanisms of male sterility-related genes are of primary importance for both basic research and application of rice breeding [2,15]. In this study, we further characterized the roles of rice *OsGPAT3* in male fertility, PCD-induced anther wall degradation, and pollen maturation on the basis of original research showing that *OsGPAT3* significantly affects anther cuticle biosynthesis and pollen exine formation [63]. This work provides new insights into the role of *OsGPAT3* in anther development and pollen formation.

### 3.1. OsGPAT3 Controls Ubisch Body Formation and Anther Development in Rice

Sporopollenin precursors are recognized as the most important structural security and nutritional sources for pollen formation [15,71]. Previous studies showed that recessive mutation of *OsGPAT3* can result in metabolism problems such as defective anther cuticle and synthesis failure of Ubisch bodies and pollen exine, which eventually cause complete male sterility [63]. Almost all components of lipid molecules decreased significantly in *osgpat3* mutant anthers [63]. Our study confirmed that *gpat3-2* exhibited a similar phenotype to *gpat3*, i.e., smooth anther surface, abnormal collapse of pollen grains (Figure 4), and amorphic Ubisch bodies (Figure 5F), finally resulting in severe degradation of all cells within the anther locule (Figure 2N,P and Figure 5H). In addition, defects in the anther wall, especially the middle and endothecium layers, were still visible at stage 13. In the *gpat3-2* mutant, this not only blocked transport of lipid substances, such as orbicules from the tapetum layer to the anther wall (Figure 2N,P and Figure 5H), but also hindered anther dehiscence. Expression changes of nutrient metabolism-related genes also confirmed that there are problems in synthesis related pathways. However, the upregulation of *MSP1* and *OsMADS3* may provide a sign that the *gpat3-2* mutant still carried active cell differentiation and degradation for sporogenesis and pollen wall formation. Different expression patterns between *OsABCG26* and OsABCG15/*PDA1* also implied that tapetum-synthesized lipid molecules differ in the process of transport from tapetum cells to anther surface by OsABCG26, and sporopollenin precursor transport to anther locules for anther cuticle formation by OsABCG15/*PDA1* [42,43,44]. These genes most likely still work to ensure cell activity in the mutant. Thus, our study demonstrated that *OsGPAT3* is required for sporopollenin precursor formation, pollen maturation, and even anther dehiscence in rice. but may functions varied in cell activity and nutrient transportation.

### 3.2. Loss of Function of OsGPAT3 Causes Abnormal PCD of Both Anther Wall and Pollen Grains

Timely cell differentiation and degradation of the three inner anther somatic layers is of critical importance for the patterning of microspore release, sporopollenin biosynthesis, pollen exine formation, and anther dehiscence. TUNEL assay analysis confirmed that *gpat3-2* exhibited delayed tapetum PCD and abnormal degradation of outer anther layers (Figure 3). The expression pattern of *OsGPAT3* in the *gpat3-2* mutant also suggests a possible feedback regulation of *OsGPAT3* transcription that functions in gradually strengthened degradation of anther wall and microspores in rice. Upregulation of tapetum PCD-related genes such as *bHLH TFs* (*UDT1*, *TDR*, *EAT1*, and *TIP2*) also showed that *gpat3-2* still had vigorous cell activity at stage 13, the appropriate anther dehiscence stage of wild type (Figure 9). However, the expression levels of *OSDEX1*, *GAMYB*, *AP25*, *AP37*, and *OsCP1* decreased significantly resulting from the upregulation of *OsGPAT3*. These results suggest that *OsGPAT3* positively regulates the expression of *OSDEX1*, *GAMYB*, *AP25*, *AP37*, and *OSCP1* in the tapetum, while *bHLH TFS*, *TGA10*, *PTC1*, and *MTR1* may be involved in other parallel regulatory pathways. Unfortunately, the functional network of *OsGPAT3* in rice tapetum development remains unclear; future investigations need to focus on identifying potential upstream genetic regulators using genetic and molecular approaches by expression pattern analysis of *OsGPAT3* in mutants of known PCD or lipid metabolism-associated transcription factors. Researchers can also create more double mutants or even polygene mutations using gene editing.

Our cytological observations showed that the *gpat3-2* mutant can release approximately normal microspores at stage 9 (Figure 1H, Figure 3D and Figure 5), which is significantly different from the *gpat3* mutant reported previously; microspores in *gpat3* were still covered with callose and could not be released from the tetrads at stage 9 [63]. Unfortunately, the newly released micropores in *gpat3-2* mutant gradually shrunk and degraded with the tapetum and other anther tissues at later stages. There were only remnants of abnormal epidermis left until the end in the *gpat3-2* anthers, all CRISPR/Cas9-induced mutants, and allelic mutants (Figure 3H,J,L and Figure 8I–N). This probably resulted from the above defects that hindered synthesis of Ubisch bodies/sporopollenin, but also abnormal PCD of the whole anther wall and pollen grains. In addition, our results also suggested that the middle layer and endothecium layer of the *gpat3-2* mutant had very little degradation (Figure 1 and Figure 5), and the PCD process of all four anther layers were delayed (Figure 3H). Although the anther wall of *gpat3-2* anthers showed accelerated degradation at late development stages, it was too late for anther dehiscence as the pollen grains were already degraded and totally aborted (Figure 2P, Figure 3L and Figure 5). Therefore, our findings demonstrate that *OsGPAT3* is also a key regulator coordinating the differentiation and degradation of different layers for normal pollen exine formation and maturation during late anther development.

### 3.3. Proposed Functions of OsGPAT3 in Rice Male Reproductive Development

Our results further confirm that *OsGPAT3* affects not only metabolic processes such as lipid and carbohydrate metabolism in the early stages of anther development [63], but also differentiation and degradation of the anther wall in rice anthers to control both anther development and pollen formation. Therefore, a putative summary describing various functions of *OsGPAT3* during the male reproductive development process in rice is given below. *OsGPAT3* directly or indirectly affects various lipid and carbohydrate metabolism-related genes to ensure proper biosynthesis of anther cuticle and glycometabolism/sporopollenin precursors for anther development, pollen exine formation, and pollen maturation. *OsGPAT3* does not, however, directly regulate the transportation pathway of these substances. It also directly or indirectly regulates expression of *GAMYB*, *DEX1*, *AP25*, *AP37*, and *OsCP1* required for tapetum development and degeneration. *GAMYB* showed downregulation probably because it also affects pollen exine/Ubisch body formation and gibberellin-inducible nutrient mobilization [26,72]. Furthermore, *OsGPAT3* directly or indirectly affects differentiation and degradation of the anther wall required for pollen maturation and anther dehiscence. In summary, *OsGPAT3* regulates processes required for proper anther development and pollen maturation in different tissues at different stages of rice anther development. Our study also increased the known novel functions of the *OsGPAT3* gene during male reproductive development in rice.

## 4. Materials and Methods

### 4.1. Mutant Material and Plant Growth Conditions

The *gpat3-2* mutant (sp. *indica*), as the pollen acceptor, was crossed with wild-type Zh8015 and 02428 (sp. *japonica*). The heterozygous F_1_ plants were then self-pollinated to generate a BC_1_F_2_ and an F_2_ population for genetic analysis and mapping of the *Osgpat3-2* gene. In the F_2_ mapping population, male-sterile plants were selected for gene mapping. All plants were grown in paddy fields of the China National Rice Research Institute during the spring of 2016 and 2017 in Lingshui, Hainan Province, China and the summer of 2016 and 2017 in Hangzhou, Zhejiang Province, China.

### 4.2. Identification of Mutant Anther and Pollen Development

Plant materials were photographed with a Nikon D800 digital camera and a Carl Zeiss SteREO Lumar V12 stereo fluorescence stereomicroscope (Markku Saari, Jena, Germany). For I_2_/KI pollen staining, anthers and pollen grains were immersed in 1.2% I_2_/KI solution (30 s) and then photographed using a Leica DM2500 microscope [12,67].

For cross-section observation of anther development, materials were collected and fixed on standard plastic sections as described by Zhang et al. and Li et al. [12,66]. Spikelets of wild type and *gpat3-2* mutant at different developmental stages were collected, based on the length of spikelet, and fixed with FAA (5% *v*/*v* formaldehyde, 5% *v*/*v* glacial acetic acid, and 50% *v*/*v* ethanol) overnight at 4 °C, dehydrated in a graded ethanol series (50–100%), embedded in Technovit 7100 resin (Heraeus Kulzer, Hanau, Germany, and polymerized at 60 °C. Transverse sections of 2.5-μm slices were obtained using a Leica RM2265 Fully Automated Rotary Microtome, stained with 0.2% toluidine blue (Chroma, Solms, Germany) and photographed using a Leica DM2000 biological microscope (Chroma, Solms, Germany).

For transmission electron microscopy (TEM), spikelets at various stages of development were collected and fixed in 2.5% glutaraldehyde in phosphate buffer (pH 7.0) for 16–24 h and then washed with phosphate-buffered saline (PBS; pH 7.2) three times, post-fixed with 1% OsO4 in phosphate buffer (pH 7.0) for 1 h, and washed three more times in phosphate buffer. Following ethanol dehydration, cutting and staining were performed as described previously [73]. After developmental identification, transverse sections were examined with a Model H-7650 TEM (HITACHI, Tokyo, Japan). For scanning electron microscopy (SEM), anthers at various developmental stages were collected and processed as described previously [67].

### 4.3. TUNEL Assay

The terminal deoxynucleotidyl transferase-mediated dUTP nick-end labeling (TUNEL) assay was performed as in a previous report [29]. Anthers from wild type and *gpat3-2* at different developmental stages were collected and prepared to generate paraffin sections. The selected paraffin sections were dewaxed in xylene and rehydrated in an ethanol series. TUNEL assay was performed using an In Vitro DeadEnd^TM^ Fluorometric TUNEL System, using fluorescein (Promega, Madison, WI 017959, USA) according to the manufacturer’s instructions with some modifications. Signals were observed and imaged under a fluorescence confocal scanner microscope (ZEISS LSM 700, Jena, Germany). All pictures were taken in the same setting.

### 4.4. Molecular Cloning of Osgpat3-2

For mapping the *Osgpat3-2* locus, total DNA was extracted from fresh leaves using the modified cetyl trimethyl ammonium bromide (CTAB) method [74]. InDel (insertion/deletion) markers within the preliminary-mapping region were developed according to sequence differences between the genome sequence of *japonica* Nipponpare and *indica* 9311 (http://www.gramene.org and http://blast.ncbi.nlm.nih.gov), and polymorphisms between the two parents (*gpat3-2* and 02428) were detected. Potential candidate genes for all open reading frames (ORFs) in the fine-mapping interval were identified by referring to the Rice Genome Annotation Project Database (http://rice.plantbiology.msu.edu/index.shtml). The primers for molecular cloning of *Osgpat3-2* are listed in Appendix A (Appendix A). The PCR products were separated by electrophoresis on 8% non-denaturing polyacrylamide gels and visualized by 0.1% AgNO_3_ staining and NaOH staining with formaldehyde.

For enzyme digestion analysis of the mutation site, the restriction endonuclease *Nla*IV site of the target gene was identified using Primer Premier 5.0 software. Two pairs of primers were designed to generate 298 bp of DNA fragments amplified by KOD-FX (TOYOBO, Osaka, Japan). Purified PCR products were used for *Nla*IV digestion. A total of 100 µL of this reaction system contained 3 µL of HaeII (20 units per 1 µL), 10 µL of NEBuffer (1×), 60 µL of purified DNA template (3 µg), and 27 µL of distilled deionized H_2_O (ddH_2_O). Two reaction systems were incubated at 37 °C for 3 h followed by 2.5% agarose gel electrophoresis.

### 4.5. Complementation of the gpat3-2 Mutant

For complementation, a genomic DNA fragment ~7908 bp containing the entire *Osgpat3-2* coding region, a 3350-bp upstream sequence, and a 1180-bp sequence downstream of the termination codon was amplified from wild-type Zhonghui8015 using the primers listed in Appendix A (Appendix A). The amplified fragment was released by *BamH*I digestion and cloned into *BamH*I-digested binary vector pCAMBIA1300 (CAMBIA, Portland, OR, USA, hygromycin resistance) using an In-Fusion Advantage Cloning kit (catalog no. PT4065; Clontech, San Francisco, CA, USA). Then, calli induced using BC_1_F_2_ seeds and showing the *gpat3-2* genotype were used for *Agrobacterium tumefaciens*-mediated transformation (all calli were selected by sequencing using the primers listed in Appendix A, Appendix A).

### 4.6. Vector Construction for CRISPR/Cas9-Mediated Mutation

Vector construction of CRISPR/Cas9-mediated mutation was processed essentially as described in Wu et al. (2017) [75]. To create the single guide RNA (sgRNA)/Cas9-induced *OsGPAT3* construct, a 23-bp *OsGPAT3*-specific sgRNA/Cas9 target sequence (red marking sequence) was inserted into the *Aar*I site of the pcas9-sgRNA vector, as described in Miao et al. (2013) [76], with some modifications. The primers used are detailed in Appendix A (Appendix A). The aforementioned calli were introduced into wild-type and Zhonghua 11 seeds by *Agrobacterium tumefaciens*-mediated transformation as described previously [77]. The T_0_ transgenic mutant plants regenerated from hygromycin-resistant calli were examined for the presence of transgenes using specific Cas-seq primers (Appendix A, Appendix A).

### 4.7. RNA Extraction, First-Strand Synthesis, and qPCR Analysis

Rice anthers at different developmental stages of the *gpat3-2* mutant and wild-type plants were collected for qPCR analysis of gene expression levels. Total RNA was extracted using the TIANGEN RNAprep Pure Plant Kit as described by the supplier. RNA was then reverse-transcribed (RT) from DNase I-treated RNA using Oligo-dT (18) primers in a 20-µL reaction using a SuperScript III Reverse Transcriptase Kit (TOYOBO, Japan). For qPCR, first-strand complementary DNA (cDNA) was diluted three times and then 3 µL of the RT products were used as the template of every PCR reaction using SYBR Premix Ex *Taq* II (TaKaRa) according to the manufacturer’s instructions. The qPCR analysis was performed on a Roche LightCycler 480 device using gene-specific primers with the rice *Actin* gene (*Os03g0234200*) as an endogenous control, the relative expression levels were measured using the 2^−Δ*C*t^ analysis method, and the results were represented as means ± SD. This analysis examined expression of *OsGPAT3* and two homologs of *OsGPAT3* (Appendix A), twelve tapetum PCD-related genes, twelve regulators that participate in lipid metabolism, and three glycometabolism-related regulators [21,31,41,44,50,78].

## 5. Conclusions

Male reproductive development in rice is important in both the improvement of yield and for an in-depth understanding of the mechanisms of anther development and pollen formation. In this study, we isolated and characterized a candidate recessive gene *OsGPAT3* that regulates anther wall PCD and pollen formation in rice using a typical map cloning method. Complementation analysis and knock-out experiments with the candidate gene further confirmed that the recessive mutation on *OsGPAT3* was responsible for the no-pollen phenotype of *gpat3-2*. Expression patterns of male sterility-related genes also demonstrated that loss of function of *OsGPAT3* caused various alterations in expression levels of both nutrient metabolism-related and tapetum PCD-related regulators, resulting in abnormal anther wall development and pollen formation. Nevertheless, the identification and observation of the *gpat3-2* mutant, together with the five allelic mutants, provided new insights into the function of *OsGPAT3* in regulating anther development and pollen formation in rice.

## Figures and Tables

**Figure 1 ijms-19-04017-f001:**
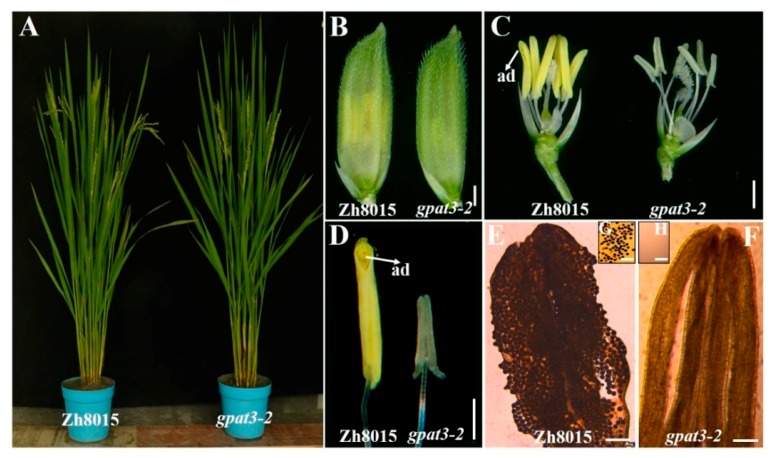
Phenotype comparison between the wild-type Zhonghui8015 (Zh8015) and *gpat3-2* mutant. Plants of the wild-type Zh8015 and *gpat3-2* mutant at the anthesis stage (**A**); Mature spikelets of the wild-type Zh8015and the *gpat3-2* mutant at anthesis (**B**); The stamen morphologies of the wild-type Zh8015and *gpat3-2* mutant; lemmas and paleae were removed for clarity (**C**); Mature anther of the wild-type Zh8015 and *gpat3-2* mutant (**D**); Compressed anthers of wild-type Zh8015 (**E**) and the *gpat3-2* mutant (**F**) after I_2_/KI staining (**E**,**F**); I_2_/KI staining of the pollen grains of wild-type Zh8015 (**G**) and the *gpat3-2* mutant (**H**) at stage 13 (**G**,**H**). Scale bars = 1 mm in (**B**–**F**), and 100 μm in (**G**,**H**); ad, anther dehiscence.

**Figure 2 ijms-19-04017-f002:**
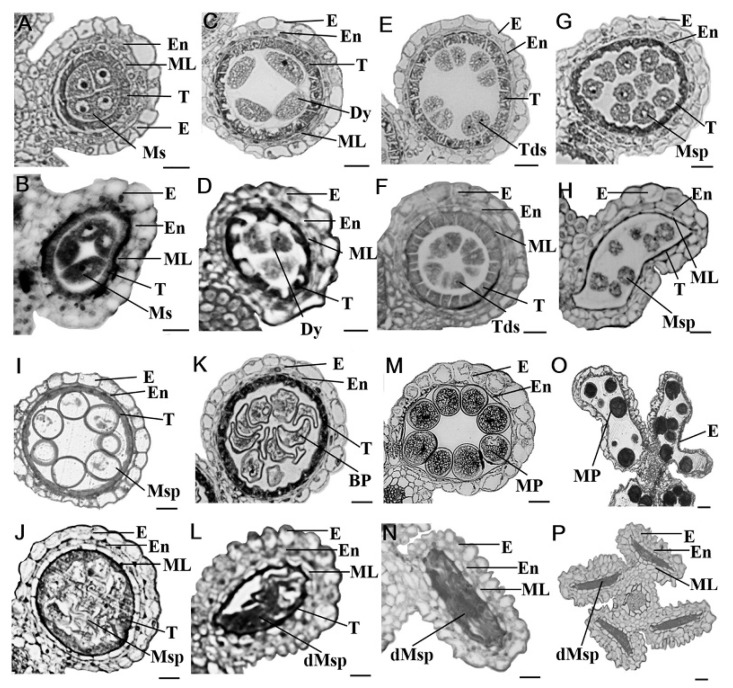
Transverse section analysis of anther development in wild-type Zh8015 and *Osgpat3-2* mutant. Locules from the anther section of Zh8015 (**A**,**C**,**E**,**G**,**I**,**K**,**M**,**O**) and *Osgpat3-2* (**B**,**D**,**F**,**H**,**J**,**L**,**N**,**P**) from stage 7 to stage 13 of development (stages 7, 8a, 8b, 9, 10, 11, 12, and 13). BP, bicellular pollen; dMsp, degraded microspores; Dy, dyed cell; E, epidermis; En, endothecium; ML, middle layer; Mp, mature pollen; Ms, microsporocyte; Msp, microspores; T, tapetum; Tds, tetrads. Scale bars = 20 μm.

**Figure 3 ijms-19-04017-f003:**
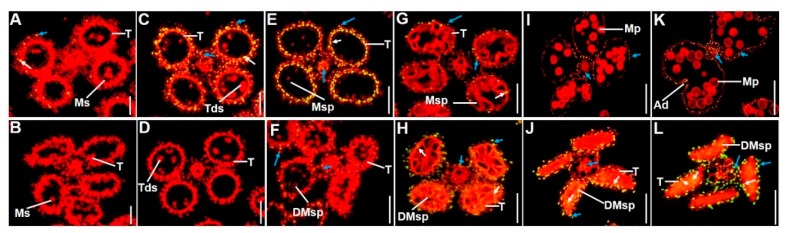
Detection of DNA fragmentation in wild-type and *gpat3-2* anthers using a terminal deoxynucleotidyl transferase-mediated deoxyuridine triphosphate (dUTP) nick-end labeling (TUNEL) assay. The wild-type and *gpat3-2* mutant anthers at stage 8a (**A**,**B**), stage 8b (**C**,**D**), stage 9 (**E**,**F**), late stage 10 (**G**,**H**), stage 12 (**I**,**J**), and dehiscence stage (**K**,**L**). A red signal indicates propidium iodide (PI) staining, while yellow and green fluorescence indicates a TUNEL-positive signal. TUNEL-positive signals detected in the tapetum cells of both wild-type and *gpat3-2* anthers are marked by white arrows, while TUNEL-positive signal observed in the outer cell layers (including the epidermis, endothecium, and middle layer), and vascular bundle cells are marked by blue arrows. Ad, anther dehiscence; DMsp, degenerated microspore; Mp, mature pollen; Ms, microsporocyte; Msp, microspore; Tds, tetrads; T, tapetum. Scale bars = 50 um.

**Figure 4 ijms-19-04017-f004:**
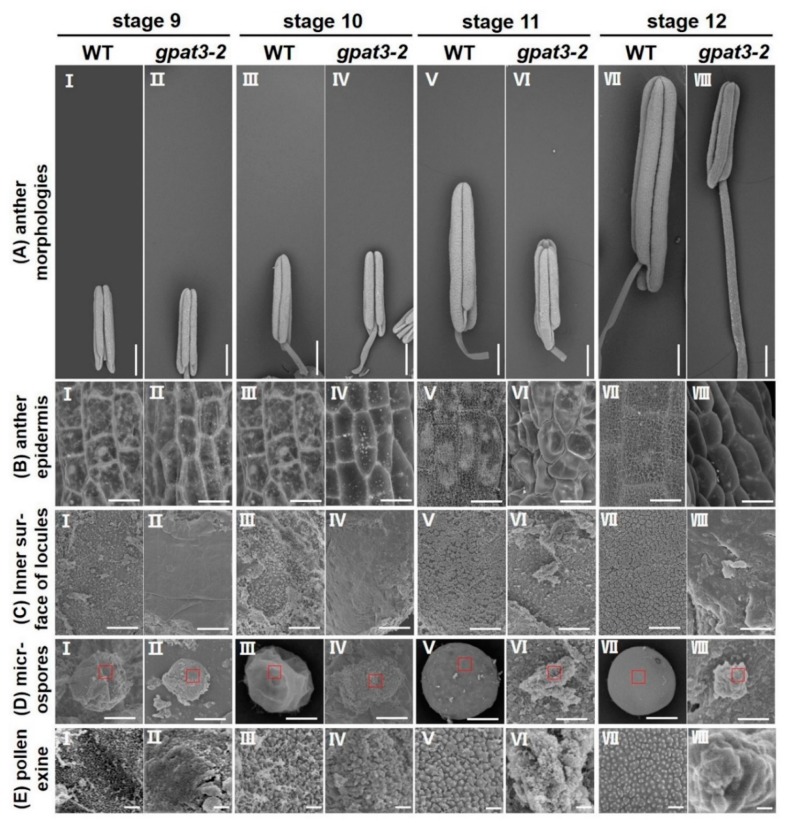
Scanning electron microscopy (SEM) observation of the surface of anther and pollen grains in wild-type (WT) and *gpat3-2* anthers. The anther morphologies (**A**), anther epidermis (**B**), inner surface of anther locules (**C**), microspores (**D**), and pollen exine (**E**) of WT and *gpat3-2* from stages 9 to 12 are shown. (**E**) Macro photograph of the outermost surface of microspores in the areas indicated by red boxes in (**D**). Scale bars = 500 μm in (**A**), 10 μm in (**B**,**C**), 5 μm in (**D**), and 200 nm in (**E**).

**Figure 5 ijms-19-04017-f005:**
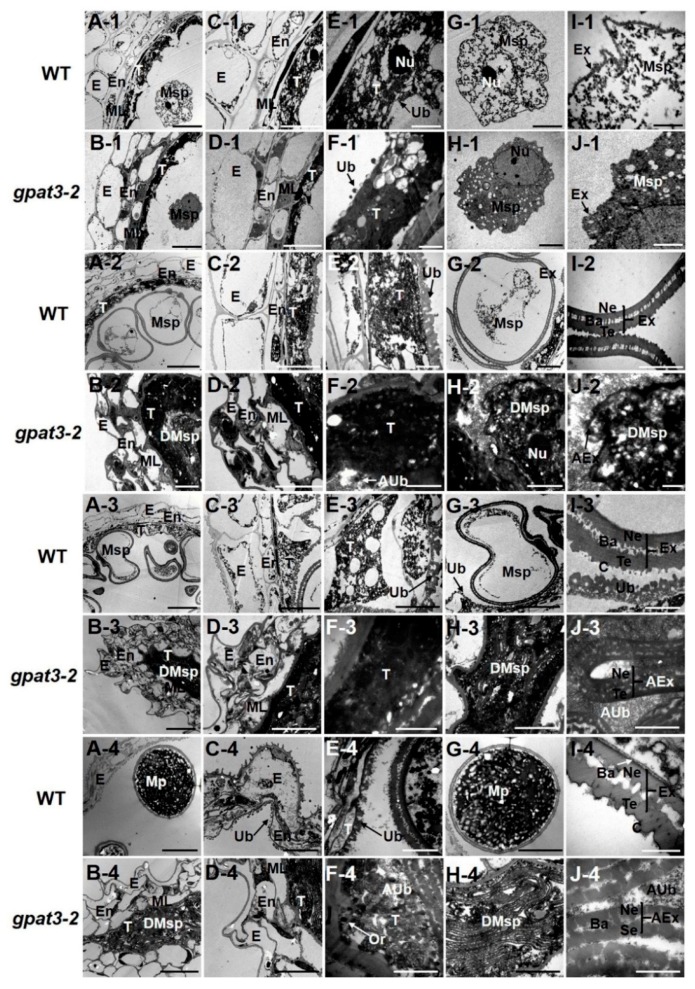
Transmission electron microscopy analysis of anther development in wild-type and *Osgpat3-2* mutant from stages 9–13. The transverse sections of the WT (**A**,**C**,**E**,**G**,**I**) and *gpat3-2* (**B**,**D**,**F**,**H**,**J**) anthers at stage 9 (**A1**–**J1**), stage 10 (**A2**–**J2**), stage 11 (**A3**–**J3**), and stage 13 (**A4**–**J4**) are compared. Anthers of the wild type (**A**) and *Osgpat3-2* (**B**), showing the anther wall with microspores at different development stages. The layers of anther wall in the wild type (**C**) and *Osgpat3-2* (**D**). Higher magnification of the tapetum cells showing Ubisch body in the wild type (**E**) and *Osgpat3-2* (**F**). The development and morphologies of microspores in the wild type (**G**) and *Osgpat3-2* (**H**). The development and structures of pollen exine in the wild type (**I**) and *Osgpat3-2* (**J**). AEx, abnormal exine; AUb, abnormal Ubisch body; Ba, bacula; C, cuticle; DMsp, degenerated microspores; E, epidermis; En, endothecium; Ex, exine; ML, middle layer; Mp, mature pollen; Msp, microspores; Nu, nucleus; Ne, nexine; Or, orbicule; Se, sexine; T, tapetum; Te, tectum; Ub, Ubisch body. Scale bars = 10 μm in (**A**,**B**), 5 μm in (**C**,**D**), 0.5 μm in (**E**,**F**), 2 μm in (**G**,**H**), and 200 nm in (**I**,**J**).

**Figure 6 ijms-19-04017-f006:**
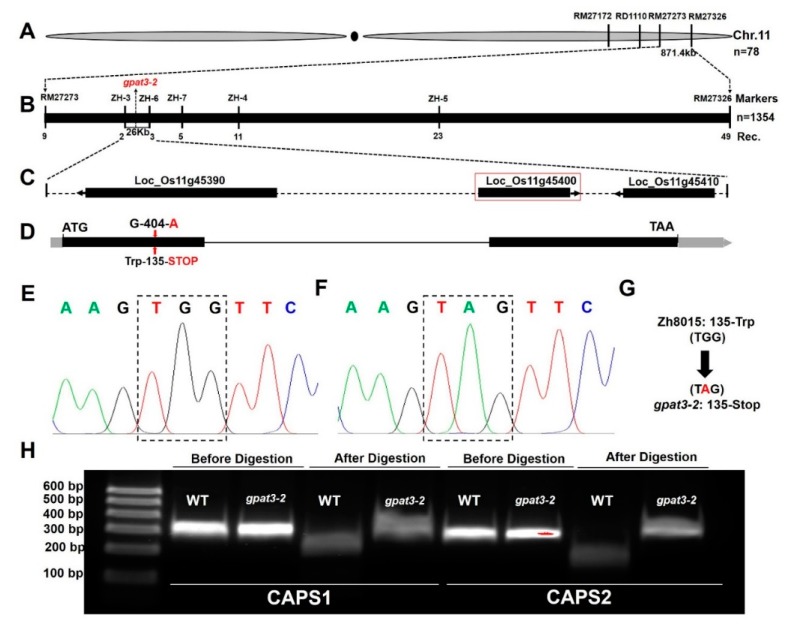
Mapping cloning of the *GPAT3-2* gene. Genetic linkage of the *gpat3-2* locus on chromosome 11 (**A**); Fine mapping of the *GPAT3-2* locus (**B**); The predicted open reading frames (ORFs) in the target region of the rice genome (**C**); Gene structure of the target gene and the *gpat3-2* mutation site (**D**); Sequences of the mutation region in the *Loc_Os11g45400* gene from Zh8015 (**E**) and the *gpat3-2* mutant (**F**); Predicted protein change of the *gpat3-2* mutation (**G**); *Nla*IV electrophoresis before and after enzyme digestion of the mutation site (**H**).

**Figure 7 ijms-19-04017-f007:**
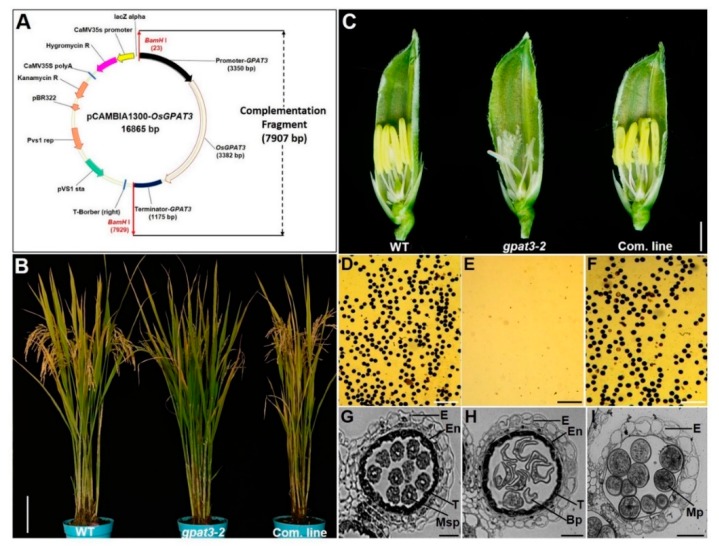
Complementation analysis of the *gpat3-2* mutant using wild-type *OsGPAT3*. The structure of the *gOsGPAT3* plasmid for transformation (**A**). The *gpat3-2* mutant was transformed with the pCAMBIA1300-*OsGPAT3* vector under the 35S promoter for transformation selection and the *OsGPAT3* genomic fragment contained three functional modules, including its native promoter, entire coding sequence, and downstream sequence from the WT for restoration of male fertility. The seed-setting rate of spikelet in wild-type, *gpat3-2*, and complementation plants (**B**). The flower of wild type, *gpat3-2*, and complementation line at stage 12; half of lemma and paleae were removed for clarity (**C**). Pollen grains of wild type (**D**), *gpat3-2* (**E**), and *gOsGPAT3*-complemented mutant (**F**) stained with 1% I_2_/KI solution at stage 12. Transverse section analysis of anthers from *gOsGPAT3*-complemented line at stage 9 (**G**), late stage 11 (**H**), and stage 12 (**I**). E, epidermis; En, endothecium; T, tapetum; Msp, microspore; BP, bicellular pollen; Mp, mature pollen. Scale bars = 15 cm in (**B**), 1 mm in (**C**), 5 μm in (**D**–**F**), and 50 μm in (**G**–**I**).

**Figure 8 ijms-19-04017-f008:**
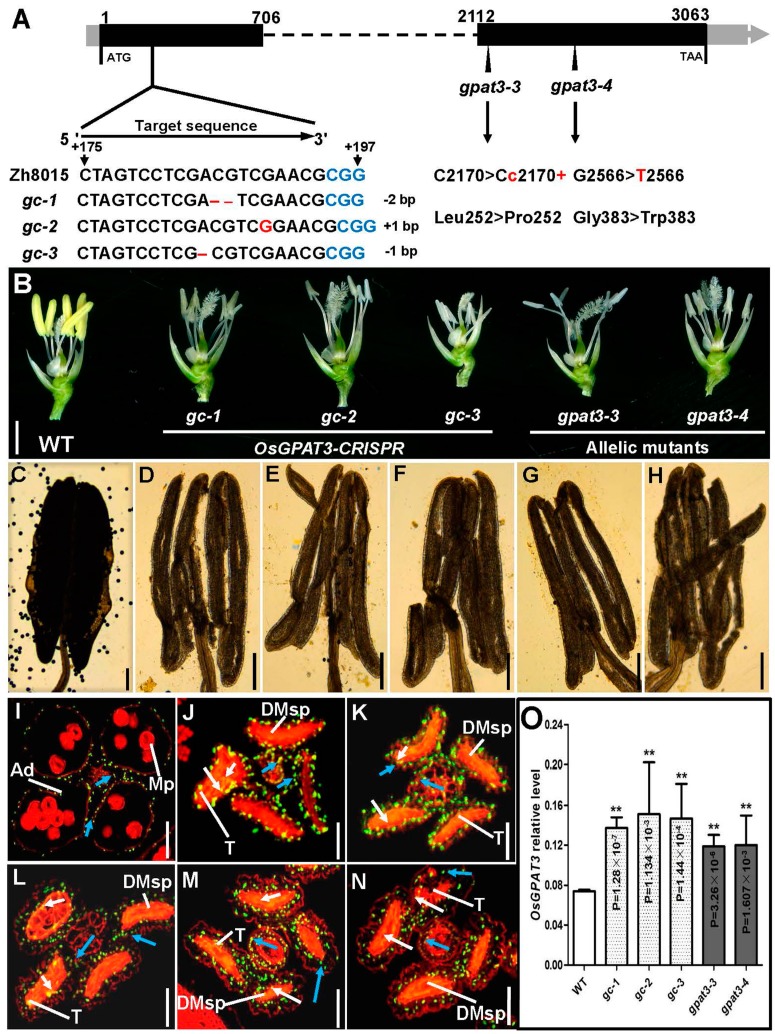
Sequence analysis and phenotypic observation of *OsGPAT3* clustered regularly interspaced short palindromic repeats (CRISPR)/CRISPR-associated 9 (Cas9)-induced mutants and allelic mutants. Gene structure of *OsGPAT3* and mutation analysis of *OsGPAT3* gene in transgenic plants and allelic mutants (**A**). The sequence (5′–CTAGTACTCGACGTCGAAGGCGG–3′) located in the first exon of the *OsGPAT3* gene was selected as the target site of single guide RNA (sgRNA). The black boxes indicate the exons. The blue characters indicate the protospacer adjacent motif (PAM). The red characters indicate the three different types of mutation events generated by CRISPR/Cas9 in the mutants. Phenotypic comparison of the WT and mutant anthers at stage 13; the lemma and paleae were removed for clarity (**B**). Compressed anthers of wild type Zh8015 (**C**) and the mutants (**D**: *gc-1*, **E**: *gc-2*, **F**: *gc-3*, **G**: *gpat3-3*, **H**: *gpat3-4*) after I_2_/KI staining. Detection of DNA fragmentation in wild-type (**I**) and mutant (**J**: *gpat3-3*, **K**: *gpat3-4*, **L**: *gc-1*, **M**: *gc-2*, **N**: *gc-3*) anthers using a TUNEL assay at the dehiscence stage (stage 13). White arrows indicate TUNEL-positive signals detected in the tapetum cells of wild-type and *gpat3-2* anthers, while blue arrows indicate TUNEL-positive signal observed in the outer cell layers (including the epidermis, endothecium, and middle layer), and vascular bundle cells. The qPCR analysis of *OsGPAT3* in wild-type, CRISPR/Cas9-induced mutants, and allelic mutants (**O**). Ad, anther dehiscence; DMsp, degenerated microspore; Mp, mature pollen; T, tapetum. Scale bars = 2 mm in (**B**), 500 μm in (**C**–**H**), and 50 μm in (**I**–**N**). ** indicates significant differences at *p* < 0.01.

**Figure 9 ijms-19-04017-f009:**
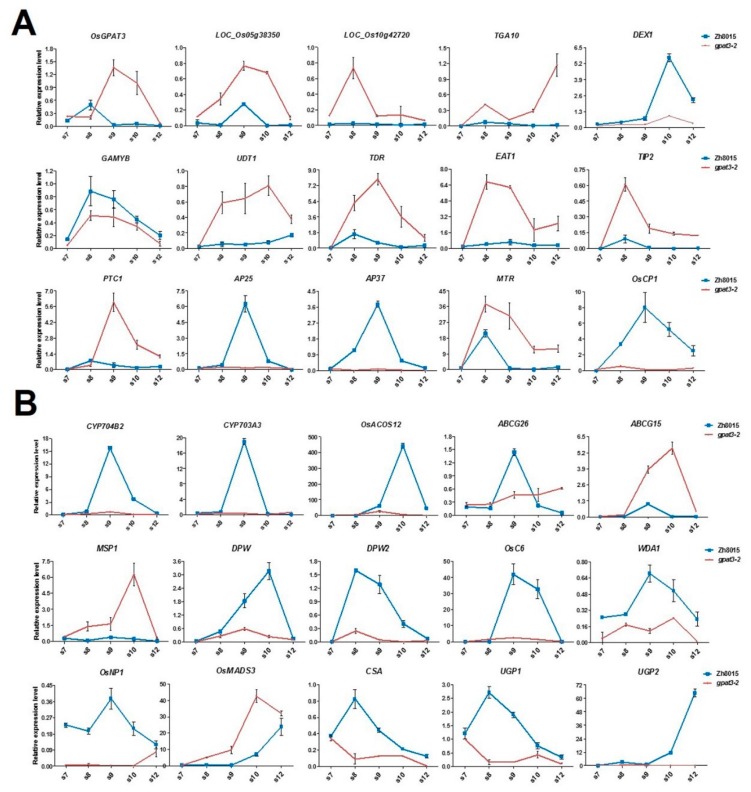
Expression analysis of *GPATs* and male sterility (MS)-involved genes in the WT and *gpat3-2*. The qPCR analysis of genes related to tapetum degeneration retardation in the wild-type and *gpat3-2* anthers at stages 7 to 12 (**A**); The qPCR analysis of genes involved in anther cutin biosynthesis/transport, sugar partitioning, and pollen exine formation in the WT and *gpat3-2* anthers at stages 7 to 12 (**B**). *OsACTIN1* was chosen as a control, and data are shown as means ± SD (*n* = 3).

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
