# Peer review of "OsGPAT3 Plays a Critical Role in Anther Wall Programmed Cell Death and Pollen Development in Rice"

_ijms, 2018, doi:10.3390/ijms19124017_

Round 1

Reviewer 1 Report

The manuscript “OsGPAT3 Plays a Critical Role in Anther Wall Programmed Cell Death and Pollen Development in Rice” by Sun et al. deals with the role of OsGPAT3 in anther development and male sterility in rice. The authors investigated development of anther wall, particularly tapetum, in the gpat3-2 mutant of rice in a great detail. They also provided functional evidence of the OsGPAT3 function by complementation and by CRISPR/Cas9 editing of the target gene. The results are valuable and shed light on male on male sterility in rice.

The main issue of the manuscript is its length, which causes difficulties in orientation for the readers. Some paragraphs are redundant.

Major comments

1. p,2,  l. 78. Ubisch bodies. Please, explain briefly, what are Ubisch bodies

2. p. 2, l. 72 “regulators” …this word is overused throughout the manuscript. Enzymes are not regulators

3. p.3, l.151 “Our study also explains the molecular mechanisms of GPATs on regulating tapetum PCD and pollen maturation during plant reproductive  development.”

This sentence is an overinterpretation. The manuscripts presents numerous numerous changes in morphology and gene expression associated with OsGPAT misfunction, but the regulatory circuits remain unknown. Please, correct throughout the text.

4. p.3, l. 156: To further understand the molecular mechanism of rice male reproductive development,…” please, exclude, similar unnecessary wording make the text too long.

5. p.5, l.189:  “the layers were epidermis, endothecium, middle layer, and tapetum,”…

Please, exclude, the layers were described previously.

6. p.6, l. 236-241. The first sentences of this subchapter belongs rather to  Introduction

7. p.7 l.253-273. Please, rewrite the description of the TUNEL essays, focusing on the most prominent results. The current description is too long and difficult to follow.

8. p. 9. TEM observations.  Please, try to combine the results of TEM with the description of transverse sections, focusing on the most important changes in anther and pollen development in the gpat3 mutant.

9. p. 13, l. 479-481. This sentence belongs to Discussion.

10. Discussion shall be completely rewritten. The first paragraph (starting in l. 536)  shall appear in Introduction. The subchapters 3.1. and 3.2.  repeat Results. It would be better to start Discussion with the paragraph 3.3.  It would be appropriate to compare the presented results with the paper by Men et al. (JXB 2017) and point to the novel findings.

11. The points stated above are just examples, how the text may be abbreviated and made more transparent. There are other cases of repetitions, which shall be eliminated.

Author Response

We would like to express our sincere thanks to you for your kindly teaching and professional review comments. We have carefully read and seriously consider these comments and made revisions according to your requirements. Our detailed response of your comments are as follows:

1.      p,2,  l. 78. Ubisch bodies. Please, explain briefly, what are Ubisch bodies

Our response: Revision is made as required.

2.  p. 2, l. 72 “regulators” …this word is overused throughout the manuscript. Enzymes are not regulators

Our response:  We have revised “regulators” to “genes”

3.  p.3, l.151 “Our study also explains the molecular mechanisms of GPATs on regulating tapetum PCD and pollen maturation during plant reproductive  development.”

This sentence is an overinterpretation. The manuscripts presents numerous numerous changes in morphology and gene expression associated with OsGPAT misfunction, but the regulatory circuits remain unknown. Please, correct throughout the text.

Our response:  Ok, thank you very much! We have change it to “our results also provide new sights of GPATs on regulating plant tapetum PCD” and corerect throughout the manuscript.

4.  p.3, l. 156: “To further understand the molecular mechanism of rice male reproductive development,…” please, exclude, similar unnecessary wording make the text too long.

Our response:  OK, we have delete this sentence and made changes throughout the manuscript.

5.  p.5, l.189:  “the layers were epidermis, endothecium, middle layer, and tapetum,”…

Please, exclude, the layers were described previously.

Our response:  OK, we have delete this part. Thank you!

6.  p.6, l. 236-241. The first sentences of this subchapter belongs rather to  Introduction

Our response:  Ok, we have remove it to Corresponding position in Introduction and the corresponding order of references throughout the manuscript have also been corrected. 

7.  p.7 l.253-273. Please, rewrite the description of the TUNEL essays, focusing on the most prominent results. The current description is too long and difficult to follow.

Our response:   We have rewrite this part. Please check that.

8.  p. 9. TEM observations.  Please, try to combine the results of TEM with the description of transverse sections, focusing on the most important changes in anther and pollen development in the gpat3 mutant.

Our response:  Thank you very much for your careful revision and suggestion. As our design, this part is one-to-one correspondence with SEM results to explain the development processes of pollen and anther cell layer, particularly on the tapetum and pollen degradation. So we think it's more reasonable to write this way.

9.  p. 13, l. 479-481. This sentence belongs to Discussion.

Our response:  Thank you! We have change it.

10.  Discussion shall be completely rewritten. The first paragraph (starting in l. 536)  shall appear in Introduction. The subchapters 3.1. and 3.2.  repeat Results. It would be better to start Discussion with the paragraph 3.3.  It would be appropriate to compare the presented results with the paper by Men et al. (JXB 2017) and point to the novel findings.

Our response:  Thank you very much for your careful revision and suggestion.  In fact, we have communicated with Dr. Men to confirm our findings. Our sights of OsGPAT3 function were as below:

1.      gpat3-2 carried the same defects of lipids synthesis, but we also find that ABCG15, MSP1, and OsMADS3 were still working in the gpat3-2 mutant. As written in 3.1.

2.      Our results confirm that gpat3-2 carried the delayed and abnormal DNA degradation in both anther wall and microspores, this is new sight of  OsGPAT3 function in regulating rice male development. As written in 3.2

3.      We made a summary of OsGPAT3 function in 3.3 to ascends our manuscript from phenotypic analysis to a network.

4.      We have done comparison with gpat3 by Men et al. (JXB 2017) and mainly focus on our new sights on PCD processes. We have rewrite this part, please check that.

11.  The points stated above are just examples, how the text may be abbreviated and made more transparent. There are other cases of repetitions, which shall be eliminated.

Our response:   We have revise the manuscript carefully and changes have been highlighted. Please check that.

 Thanks again for your kindly help and professional revision of our manuscript!

Yours sincerely,

  Lianping Sun

Reviewer 2 Report

This is a very well-written manuscript, with a good standard of writing, in terms of logic and correctness. Methods are clear, concise and accurate. Results are clearly presented. The discussion is very focused in line with the research question.

Author Response

Dear reviewer:

We would like to express our sincere thanks to you for your kind comments concerning our manuscript entitledOsGPAT3 Plays a Critical Role in Anther Wall Programmed Cell Death and Pollen Development in Rice” (No: ijms-397382). Those comments are all valuable and very helpful for revising and improving our paper, as well as the important guiding significance to our researches.

Yours sincerely,

  Lianping Sun